# Feasibility of Modified Mindfulness Training Program for Antenatal Depression and Perceived Stress Among Expectant Mothers with Male Child Preference

**DOI:** 10.3390/healthcare13060584

**Published:** 2025-03-07

**Authors:** Najma Naz, Dildar Muhammad, Khalid Rehman

**Affiliations:** 1Institute of Nursing Sciences, Khyber Medical University, Peshawar 25100, Pakistan; najma.ins@kmu.edu.pk (N.N.); dildar.ins@kmu.edu.pk (D.M.); 2Institute of Public Health and Social Sciences, Khyber Medical University, Peshawar 25100, Pakistan; drkhalid.iph@kmu.edu.pk

**Keywords:** modified mindfulness training program, antenatal depression, perceived stress, expectant mothers, male child preference

## Abstract

**Background/Objectives:** Antenatal depression and perceived stress are prevalent mental health challenges faced by pregnant women, and they are associated with male child preference. This study aimed to assess the feasibility of a modified mindfulness training program for reducing antenatal depression and perceived stress levels among expectant mothers with a male child preference at a public sector tertiary care hospital in Karachi, Pakistan. **Material and Methods:** The present feasibility trial was conducted among expectant mothers with antenatal depression and perceived stress with a male child preference using the ADAPT-ITT framework. Assessments of the needs and experiences of the new target population were carried out through an exploratory and descriptive qualitative study. In-depth interviews were conducted using a semi-structured interview guide and analyzed using a thematic analysis process. Repeated-measures MANOVA was employed to investigate the effect of time on antenatal depression and perceived stress scores in the feasibility of the intervention. **Results:** Five major themes emerged from the qualitative data. A significant influence of time was established on the antenatal depression scores, with perceived scores of F (2, 326) = 21.244, *p* < 0.001, and F (2, 326) = 310.748, *p* < 0.001. The antenatal depression scores significantly decreased from pre-intervention to post-intervention (mean difference = 4.00, *p* < 0.001), and there was a slightly significant decline from post-intervention to follow-up (mean difference = 1.167, *p* = 0.001). The perceived stress scores were significantly reduced from pre-intervention to post-intervention (mean difference = 10.214, *p* < 0.001), and there was a minor but significant decline from post-intervention to follow-up (mean difference = 0.333, *p* = 0.043). **Conclusions:** This study concludes that the modified mindfulness training program is a culturally suitable, contextually relevant intervention in the context of Pakistan and it significantly reduced antenatal depression and perceived stress in expectant mothers with a male child preference. The modified mindfulness training program was modified in accordance with the context of Islamic teaching regarding health-promoting lifestyles and religious spirituality.

## 1. Introduction

Pregnancy is considered the life event most associated with psychological and physiological change, which increases expectant mothers’ vulnerability to mental disorders [1,2]. Antenatal depression (AD) has been identified as the most prevalent psychiatric disorder during pregnancy, and can lead to dire consequences for the mother’s and infant’s health [3]. Expectant mothers are more prone to AD than non-pregnant women, with approximately twice the risk [4].

Expectant mothers with antenatal depression are at the highest risk of preeclampsia, cesarean section, a prolonged delivery process, dystocia, postpartum hemorrhage, postpartum depression, maternal suicide [5], and insomnia [6]. The children of mothers who experience AD are more likely to have a low birth weight, delayed fetal growth, and impaired brain development [7]. Furthermore, it may significantly deter their cognitive, behavioral, and emotional development [8]. AD is recognized as a global public health-related issue affecting one in five women globally [9]. The prevalence of AD reported in low- and middle-income countries is 15.6%, compared to 10% in high-income countries [10]. Additionally, the prevalence of antenatal depression was reported to be 31% in South Africa [11], 22% in India [12], 6% in Rwanda [13], and 6.8% in the USA (United States of America) [14]. In contrast, a recent study conducted in Pakistan showed the highest prevalence of AD of 82% in pregnant women [15].

Pregnancy has been identified as a stressful event in a woman’s life. Moreover, it is proven that perceived stress in pregnancy has been linked with adverse obstetric consequences, including a low birth weight, iron deficiency, and preterm birth [16]. The prevalence of perceived stress during pregnancy was 23.6% in Thailand [17], 19% in the USA [18], and 42.2% in Pakistan [19].

Higher male child preference was documented in India, at 61.23% [20], and Pakistan, at 37.5% [21]. A study conducted in Pakistan showed that male child preference is increasingly common in South Asian countries. A male child is considered an asset for specific reasons, including the male child continuing to advance the family name, caring for parents in old age, providing financial support, and ensuring safety and security to the family [22]. A research study performed in Karachi, Pakistan, exhibited a 32% male child preference rate, and a notable association was established between male child preference and antenatal stress in pregnant women [23]. Another study in Pakistan reported contributing factors to male child preference, such as parental pressure, societal norms, and pressure from in-laws [15]. Cross-sectional research conducted in Pakistan found that a preference for a male child had a value of 43.1% as a predictor for antenatal depression [24].

An MTP focuses on paying attention to the present moment and helps teach individuals with mental illnesses how to live happier and healthier lives [25]. It is demonstrated that cognitive function and performance are tremendously imperative for mental health. Cognitive function has been classified into six domains that are considered significant for overall health, encompassing social cognition, language, memory, learning, concentration, and decision-making. A healthy mind can lead to better physical and mental health. Mindfulness training has been identified as a robust method for improving cognitive health [26]. Male gender preference is one of the stressors contributing to poor mental health, primarily AD and PS [27]. Significant improvements in mental health were observed as a result of the mindfulness training program, mainly regarding AD and PS in pregnant women [28].

A suitable intervention could help pregnant women effectively manage their perceived stress levels and antenatal depression [29]. Mindfulness training program (MTP) intervention can reduce mental health problems, including stress and anxiety [30]. Even though MTPs have considerable efficacy in reducing perceived stress, longer-term follow-ups and data from various sociocultural contexts are also lacking [31]. A recent systematic review presented an MTP’s methodological quality and operationalization resilience weakness, including a small sample size and short-term measurements [32].

Pakistan is located in South Asia and has a diverse sociocultural background. There is a dearth of data regarding the feasibility of mindfulness training programs for antenatal depression and perceived stress levels among expectant mothers in the context of Pakistan. Therefore, this study aimed to assess the feasibility of a modified mindfulness training program for reducing antenatal depression and perceived stress levels among expectant mothers with a male child preference at a public sector tertiary care hospital in Karachi, Pakistan.

## 2. Materials and Methods

### 2.1. Study Design

The present feasibility trial was conducted among expectant mothers with antenatal depression and perceived stress with a male child preference using the ADAPT-ITT framework (Table 1). This study was conducted between 1 September 2024 and 31 December 2024. In addition, an exploratory and descriptive qualitative study was conducted to explore the experiences of the new target population for an assessment of needs.

### 2.2. Participants and Procedure

Eight in-depth interviews were conducted until saturation was reached using a semi-structured interview guide, and the data were analyzed using a thematic analysis process. To assess the feasibility of a modified mindfulness training program, the calculated sample size was 84 participants, with 42 subjects in the control group and 42 in the treatment group.

Pregnant women in their first and second trimesters of pregnancy, carrying single fetuses, having precious pregnancies (pregnancy after a long wait), and visiting antenatal clinics of the outpatient department at a tertiary care hospital (Sindh Govt. Qatar Hospital, Karachi) were included in this study. Expectant mothers with antenatal depression scores ≥ 10 to 30 and perceived stress scores from 1 to 40 were enrolled in this study. An antenatal depression score of ≥10 to 30 is considered the cut-off value for antenatal depression. Perceived stress is indicated by scores ranging from 1 to 40.

Pregnant mothers with a history of psychiatric illness or taking medications for psychiatric illness, including antidepressant drugs, were excluded from this study.

Before the data collection, this study’s objectives and instruments were explicitly explained to all participants. Privacy was guaranteed throughout the research process. This study was conducted following the Declaration of Helsinki of 2008.

### 2.3. Measures

The feasibility of the intervention was assessed in three phases: baseline assessment, intervention, and follow-up assessment.

#### 2.3.1. Phase I (Baseline Assessment)

In phase I, the participants were screened for antenatal depression and perceived stress due to male child preference. A convenient, non-probability sampling method was used to approach the subjects. The data were collected using an adapted, validated open-access instrument named the Edinburgh Postnatal Depression Scale (EPDS) and Perceived Stress Scale (PSS-10). The content validity index (CVI) and reliability of the EPDS were 0.78 and 0.85, respectively. The reliability of the PSS-10 was 0.84. The maximum EPDS score is 30, and a score ≥ 10 or above indicates antenatal depression. A total perceived stress score ranging from 0 to 40 was calculated. Higher scores on this scale indicate a higher level of stress; 0–13 indicates “mild stress”, 14–26 indicates “moderate stress”, and 27–40 indicates “high stress”.

#### 2.3.2. Phase II (Intervention)

In this phase, a modified mindfulness intervention was executed. A simple random sampling method was utilized to recruit 84 study participants for the feasibility of the intervention. The subjects in both the control and intervention groups were randomly allocated using computer-generated numbers. The intervention group underwent an MMTP. The control group did not undergo an MMTP and followed the routine hospital standard intervention. Initially, the PI received training on MTPs. Then, the MTP training was modified using eight vigorous phases of the ADAPT-ITT framework. The PI trained the female trainer, who carried out a plan of intervention. Weeks 5 and 6 of the MTP were modified to focus on religious spirituality and healthy eating behaviors, respectively. These changes were made based on the recommendations of the stakeholders and advisory board members during ADAPT-ITT phase 3 (Adaptation).

The ultimate objective of the modified mindfulness training program (MMTP) is to teach people with mental illnesses how to live happier, healthier lives. The six-week training includes patient-centered, evidence-based intervention focused on teaching mindfulness meditation, breath work, basic yoga, religious spirituality, and eating behavior. The two-hour intervention section was structured as follows: lecture (50 min), practice (50 min), and summary (20 min).

The outline of this modified mindfulness training program is as follows:

Week 1: The ABC model of mindful stretching and muscle relaxation was emphasized in the first week of the MMTP. The ABC mindfulness model is a short exercise that focuses on keeping grounded and calm. It is mainly utilized for controlling emotional reactions. It comprises gaining awareness about one’s thoughts and deep breathing exercises.

Week 2: The participants learned about the concept of the beginner’s mind, along with deep and attentive breathing, mindful imagery, and a mindfulness theme tune.

Week 3: The participants learned about gratitude workouts (cultivating grateful thinking) and Mindful S.T.O.P. The latter is an abbreviation for short casual mindfulness practice (S—Stop, T—Take deep or mindful breaths, O—Observe surrounding sounds, P—Proceed with the activities with a smile).

Week 4: The participants learned about body scans, kindness, and mindfulness techniques for thinking errors.

Week 5: The participants learned about cultivating a mindfulness of spirituality. This was explained using an MP3 to create a Mindfulness Personal Practice Package (Customizing Mindfulness of Spirituality Practice).

Week 6: The participants learned about healthy eating behaviors.

After six weeks of intervention, a posttest was conducted using the same questionnaire.

#### 2.3.3. Phase III (Follow-Up Assessment)

In phase III, follow-up assessment data were collected one month after the intervention from the same participants in both the control and intervention groups. The same questionnaire was used for both groups during their visits to the antenatal clinic in the outpatient department (OPD).

### 2.4. Data Analysis Procedure

In-depth interviews were conducted with eight pregnant women to assess their needs regarding the intervention. The interviews were carried out using a semi-structured interview guide until saturation was reached. Braun and Clark’s (2006) six-phase thematic analysis process was used to analyze the qualitative data [33]. This study ensured credibility, transferability, dependability, and confirmability to meet the various aspects of rigor as designated by Guba and Lincoln in 1985 [34].

To assess the feasibility of the intervention, data were entered and analyzed using IBM SPSS Statistics version 27.0. Repeated-measures MANOVA was employed to evaluate the effect of time (pre-intervention, post-intervention, and follow-up) on the antenatal depression and perceived stress scores. The multivariate Wilks’ lambda test was utilized to assess the statistically significant influence of time on the outcome variables (antenatal depression and perceived stress). A post hoc investigation using Bonferroni adjustments was conducted to compare the depression and stress scores among the three time points (pre-intervention, post-intervention, and follow-up). Before conducting the repeated-measures MANOVA, the assumptions of multivariate normality and sphericity were evaluated. The assumptions of multivariate normality were assessed using the Shapiro–Wilk test at each time point for the antenatal depression and perceived stress scores. The results indicate that both the antenatal depression and perceived stress scores met the normality assumption at three points in time (*p* > 0.05). The assumption of sphericity was tested using Mauchly’s test for both the antenatal depression and perceived stress scores, and it was not violated (*p* > 0.05). A *p*-value of 0.05 or less was considered significant.

### 2.5. Ethical Considerations

The study protocol was approved by the Graduate Committee (GC) of the Institute of Nursing Sciences (Ref No. KMU-INS/6-8/5880 on 26 March 2024), the AS&RB (Advanced Studies and Research Board) of Khyber Medical University (Ref No. DIR/KMU-AS&RB/IO/002846 on 4 September 2024), the ERC (Ethical Review Committee) of Khyber Medical University (Ref No. KMU-INS.6885 on 16 October 2024), and the ERC of Sindh Government Qatar Hospital (Ref No. SGQH/3714 on 31 October 2024). Before the data were collected, written informed consent was obtained from all study participants. Additionally, permission from the study setting for data collection was obtained from the Medical Superintendent of Sindh Qatar Hospital, Karachi.

### 2.6. Trial Registration

The trial was registered with ClinicalTrials.gov (NIH, USA) and the Clinical Trial Unit (CTU) at Khyber Medical University, Peshawar, Pakistan (NCT06685484 Unique Protocol ID: KMU/DIR/CTU/2024/009).

## 3. Results

Table 2 explores the themes and sub-themes that emerged from the exploratory and descriptive qualitative study.

The feasibility trial was carried out following the ADAPT-ITT framework. The needs assessment of the stakeholders was performed through an exploratory and descriptive qualitative study based on ADAPT-ITT phase I. The study was conducted through in-depth interviews with expectant mothers to explore their perspectives regarding intervention.

In-depth interviews were conducted with eight pregnant mothers. The sample size was based on data saturation. Five significant themes and thirteen sub-themes emerged regarding the pregnant women’s perspectives about their experiences of antenatal depression and perceived stress with male child preference and the potential acceptability and feasibility of psychological intervention. The major themes and sub-themes that appeared include the following: Theme 1: Psychosomatic response, with sub-themes such as symptoms of antennal depression and symptoms of perceived stress; Theme 2: Psychosocial determinants of maternal mental health, with sub-themes such as reproductive factors, family dynamics, financial constraints, and cultural belief and related factors; Theme 3: The influence of family dynamics on gender preference, with sub-themes such as male gender preference and the roles of in-laws and family members on gender preference; Theme 4: Coping mechanisms for managing stress and antenatal depression, with sub-themes such as faith-based coping, seeking emotional support, and relaxation through hobbies; Theme 5: Contextualized health interventions in the Pakistan context, with sub-themes such as culturally tailored approaches and religious applicable methods.

Table 3 demonstrates the respondents’ demographic and baseline clinical characteristics in the control and intervention groups regarding the feasibility of intervention. Most participants in both groups were within the age range of 25–31. No statistically significant difference was noticed in the age distribution between the groups (*p*-value = 0.468). The education level was also similar between the groups; most participants had an intermediate or middle/matric level of education (*p*-value = 0.638).

Compared to the control group, a larger percentage of participants from the intervention group were working women (control: 23.81% vs. intervention: 30.95%). However, the observed difference was statistically insignificant (*p* = 0.08). In both groups, a higher percentage of husbands had master’s level education and above, which was statistically insignificant (*p* = 0.065). Participants with lower household incomes, categorized as “Poor”, were more dominant in the intervention group, with almost statistically significant results (*p*-value = 0.058). Unplanned pregnancies were observed comparatively more often in the intervention group, whereas statistical significance was not observed (*p* = 0.08). Similarly, both groups were identical concerning the gravity and history of abortions, with statistically insignificant results.

Table 4 presents the baseline antenatal depression-related symptoms in the control and intervention groups. Participants in both groups displayed a restricted ability to laugh and delight, with more than 70% reporting lower or no pleasure (*p* = 0.081). Looking forward to pleasurable activities was equally reduced in both groups (*p*-value = 0.071). Many participants blamed themselves needlessly for things that went wrong (control: 57.1% vs. intervention: 69.1%). However, the difference was statistically insignificant (*p* = 0.636).

In the control group, 14.29% stated their inability to cope at all, in contrast to 19.05% of the intervention group. A higher percentage of the control group (21.43%) reported some difficulty in coping compared to 9.52% of the intervention group. However, statistical significance was not achieved (*p* = 0.36).

In the intervention group, more sleeping difficulties due to sadness were reported (42.86%) compared to the control group (33.3%), with no statistical significance (*p* = 0.144). Crying due to unhappiness was slightly more frequent in the control group (42.86%) than in the intervention group (33.34%) (*p*-value = 0.09).

A total of 19.05% of the intervention participants reported feeling sad “most of the time” compared to 21.43% of the control group (*p* = 0.69). Additionally, thoughts of self-harm were somewhat more frequent in the control group (45.24), in contrast to 30.95% of the intervention group, but statistical significance was not achieved (*p* = 0.055).

Table 5 compares the percentages of the responses to perceived stress in the past month between the control and intervention groups. A considerable percentage of participants from the intervention group (54.76%) demonstrated upset feelings “fairly often” due to unanticipated events compared to the control group (38.1%). However, statistical significance was not observed (*p* = 0.07). At the same time, 54.76% of the intervention group participants established an inability to control essential aspects of their life “fairly often”, compared to 38.1% in the control group (*p*-value = 0.061).

Participants belonging to the intervention group were more vulnerable to exhibiting feelings of nervousness “very often” (33.33%) compared to the control group (23.81%), but these results were statistically insignificant (*p* = 0.144). In the intervention group, 57.14% reported “rare” feelings of confidence compared to 28.57% in the control group; however, this difference was not enough to reach significance (*p* = 0.064).

While evaluating whether things were going their way, 54.76% of the intervention participants indicated “almost never”, compared to 38.1% from the control group (*p* = 0.088). Similarly, 64.29% of the participants belonging to the intervention group indicated that they had an inability to cope with things “fairly often”, while this rate was 40.48% in the control group. No statistical significance was achieved (*p* = 0.144).

A major proportion of participants in the intervention group (30.95%) stated “never” being able to overcome irritations compared to the control group (11.9%) (*p* = 0.154). Feeling “almost never” on top of things was more frequent in the intervention group (52.38%) compared to the control group (38.1%) (*p* = 0.144).

A total of 52.38% of the participants belonging to the intervention group felt anger because of things outside of their control “fairly often” compared to 38.1% in the control group (*p* = 0.254). Finally, feeling overloaded with difficulties was noted “very often” in 50% of the intervention group and 45.24% of the control group (*p* = 0.479).

Table 6 shows the baseline composite mean scores for antenatal depression and perceived stress evaluated in both the control and intervention groups. The composite mean EPDS scores in the control and intervention groups were 26.73 ± 6.42 and 28.54 ± 5.24, respectively, demonstrating a marginally higher score in the intervention group. However, the difference was statistically insignificant (*p* = 0.161). In contrast, the mean PSS scores in the control and intervention groups were 22.33 ± 2.39 and 22.85 ± 1.88, respectively, with insignificant differences (*p* = 0.268).

Table 7 presents a comparison of the antenatal depression and perceived stress scores among the participants with female and male child preferences. The mean EPDS scores were lower for the participants with a male child preference (24.71 ± 3.89) in contrast to those with a female child preference (27.17 ± 6.85), with no statistical difference between the groups (*p* = 0.171). The mean PSS scores were similar in both groups, with male preference = 22.58 ± 2.03 vs. female preference = 22.64 ± 2.66 (*p* = 0.92).

### 3.1. Post-Intervention and Follow-Up Results

Repeated-measures MANOVA was employed to investigate the effect of time (pre-intervention, post-intervention, and follow-up) on the antenatal depression and perceived stress scores. This approach was chosen as it considers the interrelated nature of the repeated observations and supports the concurrent analysis of both dependent variables (depression and stress).

Before employing the repeated-measures MANOVA, the assumptions of both multivariate normality and sphericity were evaluated. The assumptions of multivariate normality were assessed using Shapiro–Wilk tests at each time point for both the antenatal depression and perceived stress scores. The results indicate that both the antenatal depression and perceived stress scores met the normality assumption at the three time points (*p* > 0.05). The assumption of sphericity was verified by employing Mauchly’s test for both the antenatal depression and perceived stress scores, and it was not violated (*p* > 0.05).

### 3.2. Multivariate Test

The multivariate Wilks’ lambda test demonstrated a statistically significant influence of time on the variables (antenatal depression and perceived stress) = 0.201, F (4, 326) = 100.452, *p* < 0.001, indicating that the antenatal depression and perceived stress scores significantly changed across the three time points.

The multivariate interaction between time and group (intervention vs. control) was significant, with Wilks’ lambda = 0.742, F (4, 326) = 77.527, *p* < 0.001, revealing that the trend of change over time differed significantly between the groups.

### 3.3. Univariate Test

Table 8 presents the influence of time on the antenatal depression and perceived stress scores. The univariate test results indicate a statistically significant influence of time on the antenatal depression scores, F (2, 326) = 21.244, *p* < 0.001. Similarly, a significant effect of time on the perceived stress scores, F (2, 326) = 310.748, *p* < 0.001, was witnessed.

### 3.4. Post Hoc Comparisons

A post hoc investigation using Bonferroni adjustments was carried out to compare the antenatal depression and perceived stress scores for both the intervention and control groups across the three time points (T1 = pre-intervention, T2 = post-intervention, T3 = follow-up).

The antenatal depression scores significantly decreased from T1 to T2, with a mean difference of 4.00, *p* = 0.001, and from T1 to T3, with a mean difference of 5.167, *p* = 0.001. Similarly, a significant decrease was observed from T2 to T3, with a mean difference of 1.167, *p* = 0.001. The perceived stress scores significantly decreased from T1 to T2, with a mean difference of 10.214, *p* = 0.001, and from T1 to T3, with a mean difference of 10.548, *p* = 0.001. A minor but significant reduction was observed from Time 2 to Time 3, with a mean difference of 0.333, *p* = 0.043 (Table 9). The results indicate no significant changes in the depression or stress scores over the three time points (with a *p*-value greater than 0.05 for all pairwise comparisons) in the control group.

Figure 1 illustrates the mean antenatal depression and perceived stress scores in the intervention group at the pre-intervention, post-intervention, and follow-up time points.

The scatter plots indicate a significant reduction in the average scores for both antenatal depression and perceived stress over time in the intervention group. Before the intervention, the participants demonstrated higher mean scores for depression, notably inflated compared to perceived stress. After the intervention, a significant reduction in the antenatal depression scores was detected (28.548 to 24.548), while the stress scores were also reduced (22.857 to 12.643). At the follow-up, both the antenatal depression and perceived stress scores indicate a slight reduction. 

## 4. Discussion

The objective of this study was to assess the feasibility of a modified mindfulness training program on antenatal depression and perceived stress levels among expectant mothers with male child preference at a public sector tertiary care hospital in Karachi, Pakistan. The feasibility trial was conducted using the ADAPT-ITT framework. According to the ADAPT-IIT framework, an exploratory and descriptive qualitative study was conducted initially to assess the needs and experiences of the new target population.

In the present study, individuals highlighted a need for contextualized health interventions in Pakistan that align with the country’s religion, society, and culture. Individuals also underlined that the intervention must be user-friendly and have no side effects on their pregnancy. These findings are congruent with another qualitative study’s results, underscoring the need for evidence-based health interventions acceptable in the local context and religion [35].

In the present study, the results of the feasibility of the intervention in the baseline assessment and the respondents’ demographic and clinical characteristics, including age, education, employment status, household income, unplanned pregnancy, gravidity, and history of abortion, were compared in both the control and intervention groups. No significant difference in the demographic or clinical characteristics was exhibited in the baseline assessment. Hence, both groups (control and intervention) were comparable. These findings are aligned with those of a study in China by Leng et al., which found no significant difference in the baseline assessment in the control and treatment groups [36].

In the current study, the findings concerning gender preferences show that most participants reported a strong preference for a male child. However, no significant difference was observed in the control and intervention groups (*p* = 0.1). Thus, both groups are identical. These results are congruent with those of a study in China by Wang et al., which found that male child preference was prevalent in pregnant women. Furthermore, male child preference was found to be statistically significant in relation to AD [37]. A recent research study in Zambia by Tempo et al. highlighted the reasons behind male child preference, including support in old age, family security, food security, provision of financial support, and social support [38].

A study in Pakistan by Zaidi et al. showed that a male child is considered an asset for specific reasons, which contribute to the preference for sons, including the male child continuing to advance the family name, caring for parents in old age, providing financial support, and ensuring the safety and security of the family. Additionally, in societies that have a dowry-based virilocal system, male children provide assets after marriage. On the other hand, daughters are considered a financial liability because the family must prepare a dowry for her wedding [15].

In the present study, regarding baseline antenatal depression-related symptoms, the overwhelming majority (100%) of the study participants reported higher mean EPDS scores in the control (26.73) and treatment groups (28.54), even though the difference was statistically insignificant (*p* = 0.161). These study findings align with those of a study in China by Leng et al., which reported higher mean EPDS scores (10.5) in pregnant women, although statistical significance was not established. Additionally, in the trial, the control and treatment groups exhibited comparable antenatal depression symptoms at the baseline assessment [36].

Regarding perceived stress in pregnant women, the present study found greater mean PSS scores in the control group (22.33) and intervention group (22.85), and both groups were comparable in terms of the PSS scores, as statistically insignificant differences were computed (*p* = 0.268). These results align with those of a study in Denmark by Jensen et al., which revealed that the mean score on the PSS was (20) in the intervention group and the control groups, with statistically insignificant differences in both groups at baseline (*p* = 0.968) [39].

The present study found that the participants who were allocated to the intervention group of the modified mindful training program reported a significantly larger reduction in antenatal depression and perceived stress. These study findings are congruent with a study in Sweden by Lonnberg et al., which established that mindfulness-based training had a significant effect on the reduction of antenatal depression and perceived stress in pregnant women [40].

MTPs can considerably increase positive emotions, help with effective adaptation and coping with stress and depression, improve resiliency, and enhance overall well-being. MTPs are interventions that involve the mind and body in a united manner to help individuals find calm when dealing with stressors and challenges [41]. MTPs enhance body consciousness and self-regulation by maintaining sympathetic and parasympathetic responses and reducing hypothalamic–pituitary–adrenal activation. Furthermore, MTPs change psychobiological stress markers, including cortisol, C-reactive protein, and triglycerides, and can stabilize one’s mood by balancing their serotonin level [42]

In the present study, the antenatal depression scores significantly decreased from T1 (Time 1 = pre-intervention) to T2 (Time 2 = post-intervention), with a mean difference of 4.00, *p* < 0.001. Furthermore, a slight significant decrease was observed from T2 and T3 (Time 3 = follow-up). These study results are nearly congruent with those of a study in Taiwan by Pan et al. that assessed the effect of mindfulness programs on the psychological health of pregnant women, which found a lower EPDS mean score from T1 to T2 by 2.53 points at *p* = 0.007. Moreover, a significant decrease in the mean score in the EPDS between T1 and T2 was observed in the intervention group [28]. Comparable results were established in another study in Australia carried out by Sansone et al., which indicated significant differences in the EPDS mean scores in T1 and T2. In contrast, the follow-up results from T2 and T3 showed no significant differences in the mean EPDS scores (*p* = 0.130) [43]. However, a meta-analysis indicated inconsistent findings, suggesting that mindfulness-based interventions do not significantly reduce antenatal depressive symptoms in pregnant women [44].

In the present study, the perceived stress scores significantly decreased from T1 to T2, with a mean difference = 10.214, *p* < 0.001. Similarly, a slightly significant decline in the perceived stress scores was observed in T2 and T3 (*p* = 0.043). This result contrasts with those of a study in Australia conducted by Sansone et al., which did not reveal significant differences in the perceived stress scores in T1 and T2 (*p* = 0.092) or between T2 and T3 (*p* = 0.052) [32]. The variation in perceived scores may be documented using a different measurement tool, as the present study utilized the PSS-10 instrument; however, the contrasting study used the DASS-21 questionnaire.

Concerning the effect of the MMTP on AD and PS, the larger mean difference found between T1 and T2, compared to the small difference between T2 and T3, may have been due to the initial larger effect or adjustment period. However, later on, a small mean difference demonstrates a decreasing effect due to the plateau effect, where the subjects’ depression and stress scores were already lower, leading to a steadier reduction from T2 to T3 [45].

The present study results indicate no significant changes in the depression or perceived stress scores over the three time points (*p*-value greater than 0.05 for all pairwise comparisons) in the control group. These findings are congruent with those of a study in Iran by Nejad et al., who found no statistically significant results in antenatal depression and perceived scores among expect mothers in the control group [46]. The present research establishes that the MMTP is a practicable, adaptable, contextually applicable, and culturally appropriate intervention in the Pakistan context. Because this study was conducted following Islamic values, these findings are supported by recent research employed in Pakistan, which identified religious spirituality as a vital coping method in Pakistan and found a noticeable effect of religious practices on the improvement of mental health [47].

### Limitations

The present exploratory and descriptive qualitative study explored pregnant mothers’ experiences, associated factors regarding antenatal depression, and their perceived stress in relation to male child preference in the public sector in a single setting; therefore, the results may differ in private and multicenter settings.The present exploratory and descriptive qualitative study explored pregnant mothers’ experiences, associated factors regarding antenatal depression, and their perceived stress in relation to male child preference in the urban setting; therefore, the findings could be varied in rural settings.Despite the MMTP being found to significantly improve mental health in pregnant women, particularly in terms of antenatal depression and perceived stress, possible differences may have been affected by the mothers’ trimester, whether they were primiparous or multiparous, and their age group, including younger and older mothers.A feasibility trial was accomplished in one setting in a public sector in an urban area.

## 5. Conclusions

In conclusion, the modified mindfulness training program is a feasible, adaptable, contextually relevant, and culturally suitable intervention in the Pakistan context. This study also established that a modified mindfulness training program significantly reduced antenatal depression and perceived stress in pregnant women with a male child preference in the Pakistan context. This study explored pregnant mothers’ experiences of antenatal depression and perceived stress, their determinants, and how these issues are contextually and culturally construed. The results underscore the need for the development of a culturally suitable intervention in the context of Pakistan to improve the pregnant mother’s mental health. Pakistan is an Islamic country. The intervention has been aligned with Islamic values. The modified mindful training program has been revised according to the context of Islamic teachings regarding health-promoting lifestyles, including physical exercise, self-care, patience, food, and practicing religious spirituality.

## Figures and Tables

**Figure 1 healthcare-13-00584-f001:**
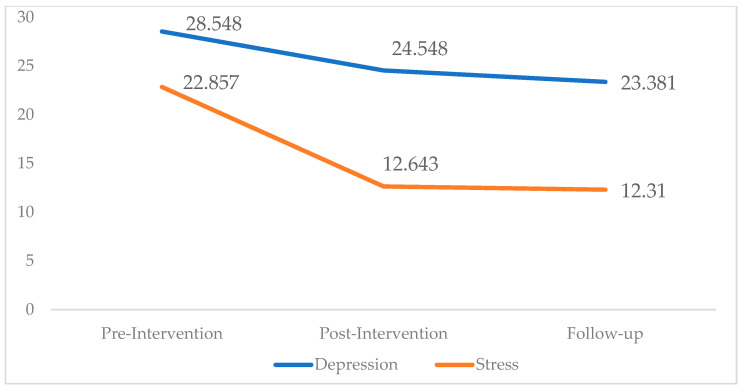
Scatterd plot of mean antenatal depression and perceived stress scores in the intervention group at the pre-intervention, post-intervention, and follow-up time points (n = 84). Legend: Antenatal depression score (28.548 to 24.548 *); perceived stress score (22.857 to 12.643 *).

**Table 1 healthcare-13-00584-t001:** ADAPT-ITT methodology.

ADAPT-ITT Phases	Methodology
1. Assessment	The community advisory board, consisting of six members, was formulated and included two patients, one nurse, one gynecologist, one psychiatrist, and one nutritionist. An assessment of needs was conducted through in-depth interviews with stakeholders such as pregnant women with antenatal depression and perceived stress.
2. Decision	Decisions related to adapting the intervention were based on the community advisory board’s recommendations and the results of in-depth interviews.
3. Adaptation	Theater testing (pretesting) of the original intervention was carried out. Furthermore, pretesting of the intervention with the target population was carried out to determine their thoughts related to the intervention, mobilize feedback on the content, and obtain recommendations for improving acceptability by target populations. Modifications were made according to the recommendations of the community advisory board; they focused on culture, ethnicity, language, cultural groups, and religions, reinforcing what is necessary to fit the country’s sociocultural context. PI demonstrated the intervention in five patients. The provider panel obtained feedback from the patients. The feedback was incorporated with the help of the community advisory team.
4. Production	The first draft of the intervention was formulated and presented before the community advisory board for further feedback and approval.
5. Topical experts	Topical experts were identified to review and provide feedback on the first draft of the intervention. The content area experts, comprising six members, were identified as two psychiatric nurses, one gynecologist, two psychiatrists, and one nutritionist. The topical experts were identified with the endorsement of the community advisory board team.
6. Integration	Feedback from the topical experts was integrated to finalize the intervention.
7. Training	Two graduate female nurses were trained to administer the intervention to the new population. The training consisted of three days of four-hour sessions.
8. Testing	A feasibility trial was conducted to assess the feasibility of the intervention. It was conducted to determine the study’s primary outcome using quantitative baseline and post-intervention data from the study participants.

**Table 2 healthcare-13-00584-t002:** Major themes and sub-themes.

Major Themes	Sub-Themes
Theme 1: Psychosomatic response	Symptoms of antenatal depression
	Symptoms of perceived stress
Theme 2: Psychosocial determinants of maternal mental health	Reproductive factors
	Family dynamics
	Financial constraints
	Cultural beliefs and related factors
Theme 3: Influence of family dynamics on gender preference	Male gender preference
	Roles of in-laws and family members on gender preference
Theme 4: Coping mechanism for managing stress and antenatal depression	Faith-based coping
	Seeking emotional support
	Relaxation through hobbies and communication with family members
Theme 5: Contextualized health interventions in the Pakistan context	Culturally tailored approaches
	Religious applicable methods

**Table 3 healthcare-13-00584-t003:** Baseline demographic and clinical characteristics of participants (n = 84).

Variable	Control n (%)	Intervention n (%)	*p*-Value
Age Group (Years)			
18–24	13 (30.95)	9 (21.42)	0.468
25–31	14 (33.33)	19 (45.23)	
≥32	15 (35.71)	14 (33.33)	
Education			
Illiterate	1 (2.38)	3 (7.14)	0.638
Primary	5 (11.90)	7 (16.66)	
Middle/matric	13 (30.95)	15 (35.71)	
Intermediate	20 (47.61)	14 (33.33)	
Master’s and above	3 (7.14)	3 (7.14)	
Employment Status			
Housewife	32 (76.19)	29 (69.05)	0.08
Working woman	10 (23.81)	13 (30.95)	
Husband Education			
Illiterate	2 (4.76)	1 (2.38)	0.065
Primary	8 (19.04)	8 (19.04)	
Middle/matric	2 (4.76)	4 (9.52)	
Intermediate	5 (11.90)	7 (16.66)	
Master’s and above	25 (59.52)	22 (52.38)	
Household Income			
Good	30 (71.42)	22 (52.38)	0.058
Poor	12 (28.57)	20 (47.61)	
Unplanned Pregnancy			
Yes	25 (59.52)	32 (76.19)	0.08
No	17 (40.47)	10 (23.80)	
Gravity			
1–2	12 (28.57)	12 (28.67)	0.497
3–4	10 (23.80)	14 (33.33)	
5–6	20 (47.61)	16 (38.09)	
History of Abortions			
Yes	11 (26.19)	6 (14.28)	0.175
No	31 (73.80)	36 (85.71)	

**Table 4 healthcare-13-00584-t004:** Baseline depression symptoms in control vs. intervention groups (n = 84).

Question	Control n (%)	Intervention n (%)	*p*-Value
I have been able to laugh and see the funny side of things			
As much as I always could	9 (21.43)	7 (16.67)	0.081
Not quite so much now	3 (7.14)	4 (9.52)	
Definitely not so much now	18 (42.86)	16 (38.10)	
Not at all	12 (28.57)	15 (35.71)	
I have looked forward with enjoyment to things			
As much as I ever did	5 (11.9)	8 (19.05)	0.071
Rather less than I used to	11 (26.19)	8 (19.05)	
Definitely less than I used to	15 (35.71)	11 (26.19)	
Hardly at all	11 (26.19)	15 (35.71)	
I have blamed myself unnecessarily when things went wrong			
Yes, most of the time	8 (19.05)	8 (19.05)	0.636
Yes, some of the time	16 (38.10)	21 (50)	
Not very often	14 (33.3)	11 (26.19)	
No, never	4 (9.52)	2 (4.76)	
I have been anxious or worried for no good reason			
No, not at all	2 (4.76)	1 (2.38)	0.36
Hardly ever	10 (23.81)	7 (16.67)	
Yes, sometimes	22 (52.38)	19 (45.24)	
Yes, very often	8 (19.05)	15 (35.71)	
I have felt scared or panicky for no very good reason			
Yes, most of the time	3 (7.14)	5 (11.9)	0.145
Yes, some of the time	17 (40.48)	10 (23.81)	
Not very often	15 (35.71)	19 (45.24)	
No, never	7 (16.67)	8 (19.05)	
Things have been getting on top of me			
Yes, most of the time, I haven’t been able to cope at all	6 (14.29)	8 (19.05)	0.221
Yes, sometimes I haven’t been coping as well as usual	9 (21.43)	4 (9.52)	
No, most of the time, I have coped quite well	12 (28.57)	9 (21.43)	
No, I have been coping as well as ever	15 (35.71)	11 (26.19)	
I have been so unhappy that I have had difficulty sleeping			
Yes, most of the time	5 (11.9)	13 (21.43)	0.144
Yes, some of the time	13 (30.95)	5 (11.9)	
Not very often	17 (40.48)	22 (52.38)	
No, never	7 (16.67)	6 (14.29)	
I have felt sad or miserable			
Yes, most of the time	9 (21.43)	8 (19.05)	0.69
Yes, some of the time	8 (19.05)	7 (16.67)	
Not very often	17 (40.48)	22 (52.38)	
No, never	8 (19.05)	5 (11.9)	
I have been so unhappy that I have been crying			
Yes, most of the time	10 (23.81)	6 (14.29)	0.09
Yes, some of the time	8 (19.05)	8 (19.05)	
Not very often	15 (35.71)	25 (59.52)	
No, never	9 (21.43)	3 (7.14)	
The thought of harming myself has occurred to me			
Yes, most of the time	8 (19.05)	5 (11.9)	0.055
Yes, some of the time	11 (26.19)	8 (19.05)	
Not very often	9 (21.43)	21 (50)	
No, never	14 (33.3)	8 (19.05)	

**Table 5 healthcare-13-00584-t005:** Comparison of perceived stress between the control and intervention groups (n = 84).

Question	Control n (%)	Intervention n (%)	*p*-Value
In the last month, how often have you been upset because of something that happened unexpectedly?			
Almost Never	0 (0)	1 (2.38)	0.07
Sometimes	13 (30.95)	4 (9.52)	
Fairly Often	16 (38.1)	23 (54.76)	
Very Often	13 (30.95)	14 (33.33)	
In the last month, how often have you felt that you were unable to control the important things in your life?			
Almost Never	0 (0)	1 (2.38)	0.061
Sometimes	13 (30.95)	4 (9.52)	
Fairly Often	16 (38.1)	23 (54.76)	
Very Often	13 (30.95)	14 (33.33)	
In the last month, how often have you felt nervous and “stressed”?			
Almost Never	1 (2.38)	1 (2.38)	0.144
Sometimes	8 (19.05)	6 (14.29)	
Fairly Often	23 (54.76)	21 (50)	
Very Often	10 (23.81)	14 (33.33)	
In the last month, how often have you felt confident about your ability to handle your problems?			
Never	14 (33.33)	9 (21.43)	
Almost Never	12 (28.57)	24 (57.14)	0.064
Sometimes	13 (30.95)	8 (19.05)	
Fairly Often	3 (7.14)	1 (2.38)	
Very Often	0 (0)	0(0)	
In the last month, how often have you felt that things were going your way?			
Never	4 (9.52)	9 (21.43)	0.088
Almost Never	16 (38.10)	23 (54.76)	
Sometimes	15 (35.71)	7 (16.67)	
Fairly Often	6 (14.29)	3 (7.14)	
Very Often	1 (2.38)	0 (0)	
In the last month, how often have you found that you could not cope with all the things that you had to do?			
Almost Never	1 (2.38)	0 (0)	0.144
Sometimes	15 (35.71)	10 (23.81)	
Fairly Often	17 (40.48)	27 (64.29)	
Very Often	9 (21.43)	5 (11.90)	
In the last month, how often have you been able to control irritations in your life?			
Never	5 (11.9)	13 (30.95)	
Almost Never	17 (40.48)	16 (38.1)	0.154
Sometimes	14 (14.29)	10 (7.14)	
Fairly Often	6 (14.29)	3 (0)	
Very Often	0 (0)	0 (0)	
How often have you felt that you were on top of things in the last month?			
Never	15 (35.71)	13 (30.95)	0.159
Almost Never	16 (38.1)	22 (52.38)	
Sometimes	11 (26.19)	5 (11.90)	
Fairly Often	0 (0)	1 (2.38)	
Very Often	0 (0)	1 (2.38)	
In the last month, how often have you been angered because of things that were outside of your control?			
Almost Never	1 (2.38)	2 (4.76)	0.254
Sometimes	20 (7.62)	8 (19.05)	
Fairly Often	16 (38.1)	22 (52.38)	
Very Often	5 (11.90)	10 (23.81)	
In the last month, how often have you felt difficulties were piling up so high that you could not overcome them?			
Almost Never	1 (2.38)	1 (2.38)	0.479
Sometimes	0 (0)	1 (2.38)	
Fairly Often	12 (28.57)	6 (14.29)	
Very Often	19 (45.24)	21 (50)	

**Table 6 healthcare-13-00584-t006:** Baseline composite mean antenatal depression and perceived stress scores in control vs. intervention groups.

Scale	Group	Mean ± SD	*p*-Value
EPDS	Control	26.73 ± 6.42	0.161
	Intervention	28.54 ± 5.24	
PSS	Control	22.33 ± 2.39	0.268
	Intervention	22.85 ± 1.88	

Legend: EPDS = Edinburgh Postnatal Depression Scale; PSS = Perceived Stress Scale; SD = Standard deviation.

**Table 7 healthcare-13-00584-t007:** Antenatal depression and perceived stress scores based on gender preference (n = 84).

Scale	Gender Preference	Mean ± SD	*p*-Value
EPDS Score	Female	27.17 ± 6.85	0.171
	Male	24.71 ± 3.89	
PSS Score	Female	22.64 ± 2.66	0.92
	Male	22.58 ± 2.03	

Legend: EPDS = Edinburgh Postnatal Depression Scale; PSS = Perceived Stress Scale; SD = Standard deviation.

**Table 8 healthcare-13-00584-t008:** Influence of time on antenatal depression and perceived stress scores, univariate tests (n = 84).

Measure	Source	Sum of Squares	df	Mean Square	F	*p*-Value
Antenatal Depression	Time	434.738	2	217.369	21.244	<0.001
Perceived Stress	Time	1771.665	2	885.837	310.748	<0.001

Legend: df = degree of freedom, F-test.

**Table 9 healthcare-13-00584-t009:** Pairwise comparisons of antenatal depression and perceived stress scores across time points in intervention and control groups (n = 84).

Group	Variable	Time (I)	Mean (I)	Mean (J)	Mean Difference (I − J)	*p*-Value
Intervention	Depression	T (1 vs. 2)	28.548	24.548	4	<0.001
		T (1 vs. 3)	28.548	23.381	5.167	<0.001
		T (2 vs. 3)	24.548	23.381	1.167	0.001
	Stress	T (1 vs. 2)	22.857	12.643	10.214	<0.001
		T (1 vs. 3)	22.857	12.31	10.548	<0.001
		T (2 vs. 3)	12.643	12.31	0.333	0.043
Control	Depression	T (1 vs. 2)	26.734	26.332	0.402	0.514
		T (1 vs. 3)	26.734	26.231	0.503	0.472
		T (2 vs. 3)	26.332	26.231	0.101	0.765
	Stress	T (1 vs. 2)	22.333	21.992	0.341	0.626
		T (1 vs. 3)	22.333	21.872	0.461	0.489
		T (2 vs. 3)	21.992	21.872	0.12	0.758

Legend: T1 = Time 1, T2 = Time 2, T3 = Time 3.

## Data Availability

The data are unavailable due to privacy and ethical restrictions.

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
