# Peer review of "Feasibility of Modified Mindfulness Training Program for Antenatal Depression and Perceived Stress Among Expectant Mothers with Male Child Preference"

_healthcare, 2025, doi:10.3390/healthcare13060584_

Round 1
Reviewer 1 Report
Comments and Suggestions for Authors
I have included my comments in the attached file.

Author Response
A point-by-point response to the reviewer’s comments
Reviewers 1
Thank you very much for taking the time to review this manuscript. Please find the detailed responses below and the corresponding revisions/corrections highlighted/in track changes in the re-submitted files
Comments 1: The study’s introduction did a great job articulating the importance of antenatal depression (AD) and the impact on maternal and infant health. However, it did not elaborate on “male preference” – the study’s key study population and cultural context. Neither did it discuss about “mindfulness training program” (MTP) and how this type of intervention is expected to be effective in addressing antenatal depression among women stressed by male preference. I suggest authors provide more background information on 1) male preference among study participants and in the context of the study, especially Karachi of Pakistan, and 2) what is MTP and why it is expected to work for women with male preference.
Response 1: Thank you for your valuable comments. I have incorporated it. Change can be found in red highlighted, on page number 02, paragraph number 3-4 and line numbers 62-83
Comments 2: The study also used “modified MTP” as the intervention. What components of the MTP have been modified?
Response 2: A lot of thanks for such an important comment. I agree it should be mentioned. MTP weeks 5 and 6 were modified by religious spirituality and healthy eating behaviour. It was modified according to the recommendation of stakeholders and advisory board members during the ADAPT-ITT phase number three (Adaptation). It can be found in red highlighted, on page number 04, paragraph number 3 and line numbers 144-147.
Comments 3: In the Abstract and Conclusion, the study pointed out “culturally suitable, contextually relevant intervention in Pakistan’s context”. However, I am less clear on how such intervention has been tailored to the Pakistan context and what does “culturally suitable” mean for this intervention. Can you give some examples of intervention components that are “culturally suitable”?
Response 3: This is one of the great comments. Pakistan is an Islamic country, in MMTP intervention week 5, expected mothers are advised to offer five times prayers in a day to attain religious spirituality. In week 6, it is advised to change behaviour by eating healthy food like protein. It is suggested to consume animal protein from Halal meat. Halal meat means it is allowed as per Islam religion. Pig meat is strictly prohibited according to the Islamic religion. Because of the aforementioned reason, it has been tailored to the Pakistan context and is culturally suitable.
Comments 4: What is the connection between in-depth interview (e.g., Table 2) and testing the feasibility of the MTP? For example, in the first paragraph under “Results”, the authors wrote 5 major themes and 13 sub-themes. However, I am less clear of how these themes were linked to the “potential acceptability and feasibility of psychological intervention”. Can you provide a paragraph and elaborate on this connection?
Response 4: This is a nice comment. It has been elaborated further in a separatee paragraph. It can be found in red highlighted, on page number 06 paragraph number 4 and line numbers 217-221.
Comments 5: The study suggested the feasibility of MMTP on reducing antenatal depression and perceived stress. What are the mechanisms of the intervention that lead to decreased depression and stress? I suggest the authors provide a paragraph discussing about potential mechanisms from this intervention.
Response 5: Thank you for highlighting such a valuable comment. I agree it. The mechanisms of the intervention that lead to decreased depression and stress have been discussed in a paragraph. It can be found in red highlighted, on page number 15, paragraph number 3 and line numbers 442-449.
Comments 6: The effect on reduced AD and perceived stress is more salient between T1 and T2 than the period between T2 and T3. I suggest the authors provide more explanation of the faded effectiveness in the Discussion. .
Response 6: Thank your comment. This comment has been addressed. It can be found in red highlighted, on page number 15, paragraph number 6 and line numbers 472-476.
Comments 7: There are missing numbers in Table 5: Column “Intervention n (%)” for the first question, there is no “n” for “sometimes”; For “how often have you felt that you were on top of things in the last month”, there is no “P value”.
Response 7: Thank you comment. It is not missing a number, 0 (0) means 0 participants rated this option. The p-value has been written. It can be found in red highlighted, on page number 11.
Comments 8: Please use 1 decimal place throughout all tables. For example, in Table 5, the authors used two decimal places, whereas in Table 3, some values are reported with 1 decimal place and others with 2.
Response 8: Thank you for your comments. 2 decimal places have been followed throughout all tables. It can be found on pages number 7-11.
Comments 9: Although not statistically significant (given small sample size), it seems that the women in the treatment group differ from those in the control group in some sociodemographic characteristics (e.g., more “intermediate” education; more working; more with poor household income, more with unplanned pregnancy) and mixed baseline depression symptoms (e.g., more in not able to “see funny side of things” and “look forward with enjoyment to things” but fewer in “crying” and “thought of harming”). Given the small sample size, I suggest the authors not emphasize too much on p-values to discuss about the differences between 2 groups of individuals. Instead, the authors can discuss about how these potential baseline differences between the two groups may contribute to the findings
Response 9: Thank you, your valuable comment. I do agree with the point. However, authors have much emphasis on p-values for knowing difference between two control and treatment groups. Because bother groups were almost identical at baseline. In post-test, difference was found in treatment group due to intervention. Significance was measured by p-values.
Reviewer 2 Report
Comments and Suggestions for Authors
-
- Please explain what are T1-T3 in abstract (T1 = Pre-Intervention, T2 = Post- Post-Intervention, T3 Follow-up) as some people might read only abstract and do not understand what are those!
- Please explain “Higher male child preference” causes and consequences in a separate paragraph. It is mentioned only in one sentence in the introduction.
- Edit” Paki-stan context” in line 70.
- Explain what does precious pregnancy in line 87 mean!
- Explain what does “antenatal scores ≥10” in line 89 mean! Scores of what?
- Explain “stress levels of 1 to 40” in line 89 of what kind of measurement?
- What is “ABCs model” in line 123?
- What is “OPD” in line 142?
- This sentence need to be revised : Need assessment for intervention, in-depth interviews were conducted using a semi-structured interview guide by eight pregnant women based on saturation.
- This sentence is missing a comma after "intervention" to ensure clarity: To assess the feasibility of the intervention data was entered and analyzed using IBM
- I think this should be corrected: The trial was registered with Clinical Trials.gov. NIH, USA and Clinical Trial Unit (CTU), Khyber Medical University, Peshawar, Pakistan. Corrected version: "The trial was registered with ClinicalTrials.gov (NIH, USA) and the Clinical Trial Unit (CTU) at Khyber Medical University, Peshawar, Pakistan."
- In figure1, it is better to indicate significant difference by * or # symbols and mention it in figure legend!
- Why Figure1 shows only one group?
- I think “The recent research study in Zambia “ could be “ A recent research study in Zambia”
- Discussion part should be revised seriously. For example: if no statistical significance was established (p = 0.193) in Nepal, it means that the study did not find strong enough evidence to support the claim that male child preference is considerable among pregnant women.
- Funding: This research received no external source of funding, is there and internal fund?
Author Response
A point-by-point response to the reviewer’s comments
Reviewers 2
Thank you very much for taking the time to review this manuscript. Please find the detailed responses below and the corresponding revisions/corrections highlighted/in track changes in the re-submitted files
Comments 1: Please explain what are T1-T3 in the abstract (T1 = Pre-Intervention, T2 = Post- Post-Intervention, T3 Follow-up) as some people might read only the abstract and do not understand what are those
Response 1: This is an effective comment. It has been written. It can be found in red highlighted, on page number 1, paragraph number 1, line number 26-30
Comments 2: Please explain the “Higher male child preference” causes and consequences in a separate paragraph. It is mentioned only in one sentence in the introduction.
Response 2: Thank you, for your comment. It has been mentioned in a separate paragraph. It can be found in red highlighted, on page number 2, paragraph number 3, line number 62-73.
Comments 3: Edit” Paki-stan context” in line 70
Response 3: Thank you for identifying the error. It has been rectified. It can be found in red highlighted, on page number 4, paragraph number 1, line number 91
Comments 4: Explain what does precious pregnancy in line 87 mean.
Response 4: Thank you for your comment. Precious pregnancy meaning has been mentioned in the bracket. It can be found in red highlighted, on page number 4, paragraph number 2, line number 111.
Comments 5: Explain what does “antenatal scores ≥10” in line 89 means ! Scores of what?
Response 5: This is one of the valid comments. Thank you. This is corrected, as an antenatal depression score ≥10 to 30 is considered a cut-off value to have antenatal depression. It can be found in red highlighted, on page number 2, paragraph number 1, line number 114-115.
Comments 6: Explain “stress levels of 1 to 40” in line 89 of what kind of measurement?
Response 6: Thank you for your comment. It has been corrected as perceived scores from 1 to 40 are considered cut-off values of having perceived stress. It can be found in red highlighted, on page number 4, paragraph number 2, line number 115-116.
Comments 7: What is “ABCs model” in line 123?
Response 7: Thank you for your comment. A description of the ABC model has been added. It can be found in red highlighted, on page number 5, paragraph number 2, line number 155-158.
Comments 8: What is “OPD” in line 142?
Response 8: Thank you for your comments. It has been written. It can be found in red highlighted, on page number 2, paragraph number 6, line number 168-169
Comments 9: This sentence needs to be revised: Need assessment for intervention, in-depth interviews were conducted using a semi-structured interview guide by eight pregnant women based on saturation
Response 9: I agree with the comment. It has been revised. It can be found in red highlighted, on page number 5, paragraph number 4, line number 176-177
Comments 10: This sentence is missing a comma after "intervention" to ensure clarity: To assess the feasibility of the intervention data was entered and analyzed using IBM
Response 10: This comment has identified important error, and a comma has been added. This can be found in red highlighted, page number 5, paragraph number 6 and line number 185.
Comments 11: I think this should be corrected: The trial was registered with ClinicalTrials.gov. NIH, USA and Clinical Trial Unit (CTU), Khyber Medical University, Peshawar, Pakistan. Corrected version: "The trial was registered with ClinicalTrials.gov (NIH, USA) and the Clinical Trial Unit (CTU) at Khyber Medical University, Peshawar, Pakistan."
Response 11: Thank your valuable suggestion. It has been modified accordingly. It can be found in red highlighted, on page number 6, paragraph number 3, line number 211-212
Comments 12: In Figure, it is better to indicate significant differences by * or # symbols and mention it in the figure legend!
Response 12: This is one of the good comments. It has been added accordingly.
It can be found in red highlighted, on page number 13, and line number 273.
Comments 13: Why Figure 1 show only one group?
Response 13: This is one of the nicest comments. In the treatment group, there was a significant reduction in the mean score of antenatal depression and perceived stress three times. Because of the significant effect, only one treatment group has been shown in Figure one.
Comments 14: I think “The recent research study in Zambia “could be “A recent research study in Zambia
Response 14: This is a valuable comment. It has been corrected accordingly. It can be found in red highlighted, on page number 14, paragraph number 4, line number 410.
Comments 15: Discussion part should be revised seriously. For example: if no statistical significance was established (p = 0.193) in Nepal, it means that the study did not find strong enough evidence to support the claim that male child preference is considerable among pregnant women
Response 15: One of the great comments. Thank you. It has been revised.
It can be found in red highlighted, on page number 14, paragraph number 4, line number 408-410.
Comments 16: Funding: This research received no external source of funding, is there an internal fund?
Response 16: Thank you for your comment. This research received neither external nor internal funds. This sentence is corrected as this research received no funding source. It can be found in red highlighted, on page number 16, paragraph number 5, line number 510.
Reviewer 3 Report
Comments and Suggestions for Authors
Dear Authors,
Reading this paper was a pleasure; it is a valuable, interesting, and well-conducted study, and I would be pleased to see it published. However, I believe that adding a few aspects could enhance the quality of the paper:
Introduction:
I felt that the introduction lacked an emphasis on the cultural context regarding the preference for male offspring in the studied population. Is there a cultural expectation for having male children? Could patients experience more severe depressive symptoms upon learning that their child is of the less preferred sex? If so, why? While there is a general reference to this preference, the cultural context is new to me, and I would appreciate more information on this aspect, as it is only mentioned later in line 72 when introducing the study's objective. Additionally, it would be useful to include a sentence in the introduction or discussion about gender disclosure practices in Pakistan. At what stage of pregnancy do expectant mothers typically learn about the sex of the child? Do all pregnant women want to know the sex during pregnancy?
Methodology:
I missed a more detailed description of the patient recruitment process. The introduction mentions a tertiary care hospital, and later we learn that participants were pregnant women in their first and second trimesters. How were these women associated with the hospital? Were they outpatients attending prenatal clinics, or were they hospitalized during recruitment? What were the inclusion and exclusion criteria? Additionally, the authors should address any factors that might have influenced the participants' emotional state and describe these in the limitations section. It would also be beneficial to reference the authors responsible for the linguistic adaptation of the PSS-10 and Edinburgh Postnatal Depression Scale (EPDS) used in the study. Furthermore, there is no information about the exact period during which the study was conducted. Moreover, the group characteristics section does not specify whether the sex of the child was known at the time of recruitment or if it was disclosed during the study, and whether this might have influenced the results.
Intervention Procedure:
The description of the intervention procedure is particularly valuable, and the intervention itself is both important and necessary.
Results and Statistical Methods:
The results and statistical methods seem appropriate, although I am not a statistics expert. The presentation of the results is clear and well-structured.
Discussion:
The discussion would benefit from a more in-depth interpretation of the findings. The comparison with results from other studies is informative, but it seems insufficient without a more detailed explanation of why these specific results were obtained. Additionally, I recommend adding a section on the study's limitations.
In summary, this paper addresses an important topic, and the research findings contribute meaningfully to the field.
I wish the authors success with the publication process!!
Author Response
A point-by-point response to the reviewer’s comments
Reviewers 3
Thank you very much for taking the time to review this manuscript. Please find the detailed responses below and the corresponding revisions/corrections highlighted/in track changes in the re-submitted files
Comments 1: Introduction:
I felt that the introduction lacked an emphasis on the cultural context regarding the preference for male offspring in the studied population. Is there a cultural expectation for having male children? Could patients experience more severe depressive symptoms upon learning that their child is of the less preferred sex? If so, why? While there is a general reference to this preference, the cultural context is new to me, and I would appreciate more information on this aspect, as it is only mentioned later in line 72 when introducing the study's objective. Additionally, it would be useful to include a sentence in the introduction or discussion about gender disclosure practices in Pakistan. At what stage of pregnancy do expectant mothers typically learn about the sex of the child? Do all pregnant women want to know the sex during pregnancy?
Response 1: Thank you for your wonderful comment. More information regarding male child preference in Pakistan context has been included in Introduction as well as discussion. There is no trend of knowing sex of the child during pregnancy. The change can be found in red highlighted, page number 2, paragraph number 3, and line number 62-72. Also, page number 14, paragraph number 5, and line number 414-420.
Comments 2: Methodology
I missed a more detailed description of the patient recruitment process. The introduction mentions a tertiary care hospital, and later we learn that participants were pregnant women in their first and second trimesters. How were these women associated with the hospital? Were they outpatients attending prenatal clinics, or were they hospitalized during recruitment? What were the inclusion and exclusion criteria? Additionally, the authors should address any factors that might have influenced the participants' emotional state and describe these in the limitations section. It would also be beneficial to reference the authors responsible for the linguistic adaptation of the PSS-10 and Edinburgh Postnatal Depression Scale (EPDS) used in the study. Furthermore, there is no information about the exact period during which the study was conducted. Moreover, the group characteristics section does not specify whether the sex of the child was known at the time of recruitment or if it was disclosed during the study, and whether this might have influenced the results
Response 2: Thank you for your comment. Pregnant women in their first and second trimesters, visiting antenatal clinics of the outpatient department at tertiary care hospital (Sindh Govt. Qatar Hospital) were included in the study. There were no any factors found influenced the participants' emotional state during intervention. The duration of the study has added. Linguistic adaptation of the PSS-10 and Edinburgh Postnatal Depression Scale (EPDS), authors faced no any obstacles during data collection. Changes can be found in red highlighted, page number 4, paragraph number 2, line number 111-112. Also, page number 3, paragraph number 2, line number 101-102.
Comments 3: Intervention Procedure:
The description of the intervention procedure is particularly valuable, and the intervention itself is both important and necessary
Response 3: Thank you for your valuable comment.
Comments 4: Results and Statistical Methods
The results and statistical methods seem appropriate, although I am not a statistics expert. The presentation of the results is clear and well-structured
Response 4: Thank you for your highly encouraging comments.
Comments 5: Discussion:
The discussion would benefit from a more in-depth interpretation of the findings. The comparison with results from other studies is informative, but it seems insufficient without a more detailed explanation of why these specific results were obtained. Additionally, I recommend adding a section on the study's limitations.
Response 5: Thank you for your comment. It has been added. It can be found in red highlighted, page number, paragraph number and line number
Round 2
Reviewer 1 Report
Comments and Suggestions for Authors
I appreciate the authors' efforts in addressing my comments. I suggest the authors integrate the response 3 - how the intervention is tailored to the Pakistan context in the main text. For example, add a few sentences for week 5 and week6 interventions to highlight the modified components and cultural relevance.
Round 2
A point-by-point response to the reviewer’s comments
Reviewers 1
I am greatly thankful for reviewing this manuscript and providing positive and constructive feedback which will definitely help to improve the manuscript. Please find the detailed responses below and the corresponding revisions.
Comments 3: In the Abstract and Conclusion, the study pointed out “culturally suitable, contextually relevant intervention in Pakistan’s context”. However, I am less clear on how such intervention has been tailored to the Pakistan context and what does “culturally suitable” mean for this intervention. Can you give some examples of intervention components that are “culturally suitable”?
Response 3: Thank you again for your valuable suggestion. It has been added to the main text. Changes can be found in red highlighted, page number 1, paragraph number 1 and line number 34-35. Also, page number 16, paragraph number 1, line number 483-489 and page number 16, paragraph number 3 and line number 513-516.